# Pactolo Bar: An Approach to Mitigate the Midas Touch Problem in Non-Conventional Interaction

**DOI:** 10.3390/s23042110

**Published:** 2023-02-13

**Authors:** Alexandre Freitas, Diego Santos, Rodrigo Lima, Carlos Gustavo Santos, Bianchi Meiguins

**Affiliations:** Computer Science Postgraduate Program, Federal University of Pará, Belém 66075-110, Brazil

**Keywords:** Head Tracking, Midas Touch problem, non-conventional interactions, human–computer interaction, graphical user interface, user experience

## Abstract

New ways of interacting with computers is driving research, which is motivated mainly by the different types of user profiles. Referred to as non-conventional interactions, these are found with the use of hands, voice, head, mouth, and feet, etc. and these interactions occur in scenarios where the use of mouse and keyboard would be difficult. A constant challenge in the adoption of new forms of interaction, based on the movement of pointers and the selection of interface components, is the Midas Touch (MT) problem, defined as the involuntary action of selection by the user when interacting with the computer system, causing unwanted actions and harming the user experience during the usage process. Thus, this article aims to mitigate the TM problem in interaction with web pages using a solution centered on the Head Tracking (HT) technique. For this purpose, a component in the form of a Bar was developed and inserted on the left side of the web page, called the Pactolo Bar (PB), in order to enable or disable the clicking event during the interaction process. As a way of analyzing the effectiveness of PB in relation to TM, two stages of tests were carried out based on the collaboration of voluntary participants. The first step aims to find the data that would lead to the best configuration of the BP, while the second step aims to carry out a comparative analysis between the PB solution and the eViacam software, whose use is also focused on the HT technique. The results obtained from the use of PB were considered promising, since the analysis of quantitative data points to a significant prevention of involuntary clicks in the iteration interface and the analysis of qualitative data showed the development of a better user experience due to the ease of use, which can be noticed in elements such as the PB size, the triggering mechanism, and its positioning in the graphical interface. This study benefits in the context of the user experience, because, when using non-conventional interactions, basic items such as aspects of the graphic elements, and interaction events raise new studies that seek to mitigate the problem of the Midas Touch.

## 1. Introduction

Users in technological environments are increasingly diversified, for example, in augmented and virtual reality [1], 3D projections [2], and interactive public displays [3]. For these new scenarios, users have presented difficulties in interacting using traditional devices, such as mouse and keyboard, especially users with upper limb motor limitations. In this context, non-conventional interactions give rise to technologies that allow users to interact with computer systems in other ways, such as voice commands [4], gestures commands [5], cursor movements with head movements [6] or eyes [7], and brain–computer interfaces [8], among others.

Mouse usage on actions in computer systems has two primary modes of interaction: movement and selection. Pointer movement can use several technologies, and Head Tracking (HT) stands out. HT is an alternative to low-cost non-conventional interaction compared to other technologies such as eye tracking [9]. Among the available methods for performing the selection action, the Dwell Time resource [10] is one of the primary ones, which specifies a waiting time on an interface component to perform the select action (simple click).

Including non-conventional interactions in computational systems presents several problems, including the Midas Touch (MT) problem [11]. These scenarios present their own MT conditions when adding the interaction techniques [12,13,14,15]. Some strategies are employed to mitigate this problem: Dwell Time modification, blink detection to perform [6] actions, voice commands [4], gesture commands [5], and specific graphic interfaces [16]. However, after analysis, we found that these strategies still do not present an effective way to solve the problem. This happens frequently on a particular computer system.

However, for this study, the focus will be on HT. This technology became popular through software freely available on the web, such as CameraMouse [17] and eViacam [18]. This type of software estimates the rotation of the user’s head through computer vision techniques in video frames captured by a camera for cursor movement. For the selection event, Dwell Time was used, which still presents the challenge of involuntary selection actions, in part due to the incorrect setting of the action time. This problem causes a decrease in the quality of the user experience when interacting with the computer system.

Thus, this paper proposes a strategy and evaluation method to minimize the TM problem, using HT in content presented by an Internet browser, such as web pages, video and audio players, and games. The solution provides a sidebar in the interface, with which the user can activate and deactivate the selection event (simple click), called a Pactolo Bar (PB). In general, a PB should allow ease of interaction, with less physical effort for the task and fewer involuntary selection actions when compared to other solutions, such as eViaCam.

This paper assesses PB using a two-stage method. The first stage of tests evaluate simple tasks to choose a bar option in the interface. The results of the first stage identify the best position and width to use the PB. The second stage evaluates a margin that separates the PB from the page content. In the second stage of the tests, inspired scenarios simulate real sites, where each participant performs tasks to enable and disable the selection action and select or enter an interface component. In the second stage, the analysis aims to compare the PB and the solution presented by eViacam, aiming to analyze quantitative and qualitative data of the interaction and the users’ experience concerning usability [19,20].

The name “Pactolo Bar” was inspired by the story of King Midas [21]. A god cursed King Midas, and everything he touched turned into gold. Midas, seeking redemption for this curse, asked the god what he should do to be free of it. The god answered that Midas was to go to the Pactolus River and wash his hands to get rid of the curse. Analogous to the river, PB is a proposal to correct the curse of MT (involuntary selections) in computer systems.

This article is organized as follows: "Theoretical Foundation and Related Works" describes the main concepts related to HT and Dwell Time for interaction and strategies to deal with the MT problem and presentats of the proposal of this article, as well as the studies that influenced the development of this study; "Evaluation Methodology" describes the devices used in the research, the profile of the participants, and the procedures of the tests performed; "Results" and "Discussion" present and discuss of the results found, respectively; and Future Works presents the contributions of this research to the area and proposals for further advances from the findings.

## 2. Theoretical Background and Related Works

The HT for navigation interaction in computer systems is already a popular method [22,23,24]. The work of Al-Rahayfeh et al. [25] states that the movement of the head is natural and uncomplicated and is effective for performing the action of pointing at objects. There are different ways to estimate the position of the head, including the detection of head movement from computer vision [26], methods that use acoustic signals [27], methods based on gyro sensors and accelerometers [28], and hybrid techniques [29]. Low-cost resource usage is common, and this work adopted free software solutions based on conventional cameras, such as eViacam and CameraMouse.

A common method to trigger selections with HT is the Dwell Time [10,30], in which a pre-programmed action (e.g., a single click, double click, drag, and drop) is triggered after a cursor remains over the target interface component for a set amount of time. The authors further define three types of Dwell Time: Continuous, Cumulative, and Adaptive. The Continuous Dwell Time resets the timer if the interaction is not carried until the end, while the Cumulative does not. The Adaptive Dwell Time promotes the flexibility of the parameters to establish the best timer configuration of the Dwell Time according to the particularities of each interaction.

However, since it is not trivial to differentiate if a user is deliberately performing a Dwell Time interaction over a target element or is simply in a stationary head position over an arbitrary element, the use of Dwell Time can cause involuntary actions. Such involuntary actions in interaction methods are known as Midas Touch (MT) [11]. According to [31], experiments with inertial sensors coupled to a Head-Mounted Display for typing tasks on virtual keyboards using Dwell Time indicate that timers smaller than 400 milliseconds are not efficient, due to the high occurrence of Midas Touch.

### Strategy for the Midas Touch Problem and Our Proposal

Possible solutions to mitigate the MT problem in pointer navigation in technologies such as head, face, or eye trackers [6,10] include the use of triggers or external devices [28,32,33], such as soft-switches [34], smartphones [35,36], sensors [37], commands by specific gestures [38,39,40], or body movements (head, hands, feet, tracking or blinking, or jaw movement, among others) [41,42,43,44]. Besides traditional desktop settings, some studies have also investigated solutions aimed at mobile devices [35,36], and virtual reality environments [43,45].

Most studies analyze the problem of Midas Touch through the lens of data entry tasks, such as inputting data in specific keyboard layouts [44,46]. Only a few studies focus on other tasks, such as general-purpose interaction in 2D and 3D environments [34,37], interaction in mobile device applications [47], and wheelchair movement control [33]. As for interaction details, most studies use the single click as the event for Dwell Time, with only one article considering double clicking [35].

Most of the works in the analysis considered the Dwell Time as a technique to study since it is considered fast to implement and low in cost [10,35]. Some works presented a type of adaptive Dwell Time according to the type of interaction to be used by the user [30,34,45,47,48,49]. The results of these studies show the use of Dwell Time as being close to the best results of the other techniques, but they also show that the correct timer configuration is essential to obtain good results from Dwell Time.

Thus, this work proposes the Pactolo Bar (PB) as a method to mitigate the Midas Touch with Dwell Time. The PB aims to provide the user with an easy interaction to activate or deactivate selection events. To interact with the PB, the user navigates with the cursor to contact (Hover) the PB element. The PB element changes color to indicate if the selections are enabled (blue bar) or disabled (red bar). The goal of the PB is to provide a way of disabling interaction without having to use the Dwell Time itself in an interface element.

Figure 1 shows a representation of PB (in blue) on any web page, with adjustable size/width. A blank safety bar, called the Margin Bar (MB), was also included. The other components are representations of content that are susceptible to selection action on a web page.

## 3. Evaluation Methodology

PB tests produce interaction data for usability analysis [50,51] concerning the MT problem. The evaluation has two stages: the first stage collected quantitative data on the best configuration of the BP (location and size), and the second stage generated quantitative and qualitative data to carry out the comparative analysis between the PB and the eViacam tool, observed as one of the primary solutions cited in scientific articles for the context of this study, in addition to being easily accessible, free, and open source.

### 3.1. Environment Configuration

The tests were carried out in an air-conditioned room, with controlled lighting, and an adjustable chair for the participants’ height and arms. As equipment, we used a 21-inch monitor with a resolution of 1920x1080 pixels, with an approximate distance of 70 cm from the participant, a computer with 16GB of RAM, an Intel Core i5 processor, and a webcam with support for Full HD resolutions of 1920x1080 pixels with a baud rate of 30 FPS.

The PB is a sidebar positioned on the left side of the display, and it can be enabled or disabled through the Hover action on the component. The eViacam software uses HT to control the pointer. It presents a fixed bar at the top of the display with graphic components that symbolize the changes in cursor events, simulating mouse actions.

Figure 2 shows a representation of the eViacam function bar, where the component that enables/disables the simple click is on the red square, and to perform the selected action by clicking on the component, it is necessary to use the Dwell Time feature, which for the analysis of this research was set by default to 15 tenths of a second.

### 3.2. Profiles of Participants

Different groups of volunteers participated in each test stage, a preset configuration in which the same volunteer could not participate in both stages. All participants had the option of signing the protocol and signed a consent form (PSCF). This document explains the context, presents the research objective and the purpose of using the collected data. If the volunteer signs it they are be able to participate in the tests.

For the first stage of testing, a group of 20 participants aged between 18 and 40 was formed [52]. All participants reported that they use computers daily and have had contact with web browsing. Of this first group, 8 participants stated that they did not know non-conventional interaction techniques. The others had already had contact with this type of technology. Among the 12 with knowledge about non-conventional interactions, five did not use the HT resource in computer systems at all. The other seven had already experienced it by carrying out specific tests in different studies but without elements that could compromise the data of this study. Table 1 below shows in detail the profiles of participants in this first stage.

For the second stage of tests, two groups of 12 participants each were formed [52], and the same participant could not be in both groups. Thus, 12 performed the test with the BP, and the other 12 performed the test using the eViacam software. As in the first stage, Table 2 shows the main characteristics of the participants in this test. The participants included 21 who claimed to never have used HT, and the other 3 had knowledge about the technology, although this knowledge did not affect the analysis of the research.

The definition of the groups was based on the use of computers on a daily basis, since in the second stage of tests, the scenarios were based on known systems when using internet applications, with tasks considered to be commonly carried out. These groups of the second test stage were defined by random criteria, as it is not desired that the properties of each group be different [52]. Thus, the groups are approximately comparable, and the experimental effect is observed by using the HT and the techniques assigned to mitigate the Midas Touch problem.

### 3.3. Test Procedures

Two stages of tests were used to evaluate the PB in this study. The first stage sought to find the best configuration of the PB for use in the web context. The second stage, on the other hand, was used to perform a comparative, quantitative, and qualitative analysis of PB and the eViacam software. Simulation of real-world scenarios is the basis for the comparison process through a sequence of task evaluations.

### 3.4. User Training

Before the tests, each participant underwent training on the scenario environment. This training introduced basic concepts involved in the research, such as unconventional interactions, HT, and MT. The training consisted of the participant calibrating HT technology in the following features: speed of movement of the pointer, the time established to perform the click with the Dwell Time, recognition of the PB or eViacam functionalities in the computer system, and the type of task performed.

### 3.5. First Stage of Tests

The first stage of testing consisted of random scenarios where the following elements were randomly arranged: number of bars on the screen, the position of PB and MB on the screen, the orientation of PB and MB on the screen, and the size of PB and MB. The values used in each of the elements are shown in Table 3.

The task consisted of the user choosing one of the bars to interact with, which consisted of only the act of positioning the cursor over one of the bars presented.

Figure 3 presents some of the random scenarios, with a total of 100 samples of screens for each of the 20 participants. In each sample, the test modified the following elements: horizontal/vertical size of the PB (blue) and MB (white) that separates the content from the page (light pink).

Figure 4 presents the results obtained after the first test stage, where each bar in the chart represents the interface sides of the PB. The same chart also shows the summation of participants’ interaction times, based on the choices in the random scenarios.

The Figure also clearly shows us that the PB located to the left of the interface was the one that obtained the best evaluation by the participants to perform the interaction procedures. The position at the bottom of the interface received the worst feedback.

To understand the best setting for positioning to carry out interactions with the PB, we grouped result data on the basis of each participant. As a result, values indicate the average preference for each of the BP positions as follows: Left (37.6%), Top (24.75%), Right (24.15%), and Bottom (13.5%). Using the ANOVA [53] test, it was possible to observe that these mean values showed significance in their differences (F = 36.620; *p* < 0.001).

Thus, after analyzing the results obtained, there was a need to insert another PB in the interface, and the order was as follows: Left > Right/Top > Bottom. MB is a strategy adopted to avoid more involuntary clicks on content close to PB. The MB uses the ratio of the percentage (5%, 10%, and 15%) of the screen dimensions to define the horizontal or vertical width of the PB.

Regarding the MB configuration, the vertical or horizontal width has values between 1, 2, and 4 pixels. In Figure 5, the PB at 5% of the screen width and the MB of the width of 4 pixels represented by the chart bar in orange is the best configuration.

### 3.6. Second Stage of Tests

The settings used for the PB in this stage were based on the results obtained in the first stage of tests. The vertical/horizontal scaling of the PB was configured with 5% of the display screen size and the MB having its width configured with the size of 4 pixels.

Before describing the scenarios developed for the second test stage, it is necessary to clarify that the eViacam software provides several graphic components that represent the different cursor events (single click, double click, among others). However, it is noteworthy that in this work the primary objective is to mitigate the MT problem with a specific focus directed to the issue of involuntary selection actions (single click). Thus, for eViacam, only actions involving the selection action were considered.

The developed scenarios aim to analyze the use of BP in common (frequent) tasks in the system, such as selecting one or more elements and navigating through menu structures, among others. These tasks were performed by the participants within specific environments developed for this research. For the second stage of testing, scenarios were developed that simulate real websites with the following functionalities: Video Platform [54], InfoVis Dashboard [55], Wiki [56], and E-commerce [57].

To carry out the second stage of tests, a configuration of two groups composed of 12 participants each was used, where each participant could only perform the tasks related to one tool (BP or eViacam). The first group performed the proposed tasks with BP and the second group with the eViacam software.

For the second stage of testing, each participant performed a total of four tasks in the developed scenarios. To carry out the tests, all participants initially received training on the basic concepts involved in the study and on the scenarios where the tasks would be performed.

Aiming at standardization for all participants, each task started with the cursor positioned in the center of the screen, which aimed to establish a fair comparison between the strategies addressed in the tests.

The tasks were designed to approximate real-world scenarios as much as possible, which would make the user experience more empirical. For a better organization of the analyses of this research, for each task, a circuit was defined with a minimum number of clicks to complete the respective task.

The purpose of the defined circuit is to generate data that would lead to the analysis of how each participant performed using the BP or the eViacam software, whether the participants would be above, below, or within the stipulated average for the circuit of fulfillment of tasks.

However, even though the perfect circuit was defined for carrying out each of the tasks, each participant was instructed to complete the task by seeking a specific objective, being able to explore the interface in the way they wanted to achieve their objective.

This definition of conduction is that the test can bring the spontaneity of interaction with the computational system, jointly providing the possibility of carrying out exploratory analyses of the HT+PB set. Each of the tasks and proposed scenarios are described below.

#### 3.6.1. Scenario 1: Based on the Infovis Dashboard

This scenario was developed based on a generic Information Visualization (InfoVis) [58] dashboard, bringing an interface that uses the BP with the HT, together with clustered and centralized components, and buttons in the corner of the interface, highlighting the use of a map, similar to the InfoVis technique used in this scenario.

The interacting components on the map are close to each other, which can cause involuntary clicks on elements that are not part of the objective. As it is a map, the task is to interact with the regions, highlighting components of varying sizes. In addition to the buttons in the corner of the screen, where the distance and proximity can generate the same problem in question.

The tooltip on the map has an animation where it is possible to see the extra information of each region and which, depending on the time that the participant uses to analyze the components of the proposed scenario, may cause involuntary clicks on them, a fact that brings the possibility of MT analysis more clearly and that occurs routinely in various environments of websites in the real world. Figure 6 shows the constructed scenario, simulating an infovis dashboard, with the PB present in the interface, a central map, and a component in the upper right corner to interact with.

This scenario was divided into two stages: a visual analysis, referring to the area of the regions (the “geographic view” component), and an analytical view of the map, referring to the amount of population in the region (“population view” component). The interaction with the components changes the color of the map, so it becomes a heat map referring to the respective information of the components.

In the first stage of this scenario, the participant must select the three regions that have the largest geographic area, and, for this, they receive the help of a bar graph that appears when interacting with the “Geographic View” button and the heatmap that is coded on the map using the color scale that goes from red (largest area) to gray (smallest area). Figure 7 represents the final state of the first step of the task. Thus, the sequence necessary for the participant to complete this step is presented below.

Interact with the “Geographic View” button;Interact with PB (disable click);Analyze both the bar graph and the map;Interact with PB (activate click);Select the three regions with the largest geographic areas.

In the second step, the participant needs to select the three states with the highest demographic density, and this step starts with the selection of the “Population View” button on the interface, at which point the participant receives assistance from the appearance of a bar graph and of the heatmap that is coded on the map using the color scale that goes from red (highest population density) to gray (lowest population density). Figure 8 represents the final state of the second stage of the task. Thus, the sequence necessary for the participant to complete this step is presented below.

Interact with the “Population View” button;Interact with PB (disable click);Analyze both the bar graph and the map;Interact with PB (activate click);Select the three regions with the highest population density.

#### 3.6.2. Scenario 2: Based on Video Platform

The second test scenario was developed to be a generic video platform website. This was chosen because it is an environment where there is a greater possibility of executing involuntary clicks (MT), mainly when the video player is in full-screen mode, because in this configuration the entire interface becomes susceptible to clicks that can cause unwanted actions, and the purpose of the task performed in this scenario is to watch a video in full-screen mode.

The buttons for the Play and Volume functions, in addition to being located at the bottom-left of the interface, when accessed, can cause involuntary clicks. In addition, the Volume function button has an animation, where the waiting time for activation can cause the MT problem.

In the case of the Full Screen function button, it was initially more centralized in the interface, and to access it, it is necessary to move over several other components, which may cause, once again, a real possibility of an involuntary click on any of the other interface components.

Figure 9 shows a comparison between the real scenario and the scenario developed for the tests. The sequence required for the participant to complete this step is presented below.

The purpose of the task in the second scenario is for the participant to be able to watch a video by interacting with some buttons on the interface. Figure 10 shows the final state of the task in the second scenario. The sequence necessary for the participant to complete this step is presented below.

Click on the Full Screen button;Click on the play button;Watch until about halfway through the video;Decrease the audio volume of the video by half;Interact with PB (disable click);Interact with the second PB (invisible cursor)Watch the video until the end.

An important aspect to be highlighted was the development of the second PB (Element A in Figure 10) located at the top of the video player. This second PB has the function of making the cursor invisible and preventing any accidental interaction on the interface. The joint use of the two Pactolo Bars simulates exactly what happens in the real environment, as the cursor remains invisible when it remains stationary in the interface and, therefore, hides the information during the execution of the video.

The interaction with the auxiliary PB occurs in the same way as with the main PB, with its activation carried out from the Hover action on the component. The blue color represents that the cursor is visible in the interface, and the red color indicates that it is invisible.

#### 3.6.3. Scenario 3: Based on the Wiki Page

This scenario was developed as a Wiki website due to its structure, where many elements clustered in small areas can be observed, a configuration that may favor the occurrence of MT. It is important to point out that some of the hyperlinks present the animation structure in the form of tooltips, to provide the on-demand detail feature of the accessed content. The content for the page was extracted from a real Wiki website [59].

Interaction with scrolling is necessary on this site, since, in many surveys, the desired information is not found at the top of the page, requiring the user to scroll through the interface one or more times to see the rest of the page’s content. The hyperlinked text element is widely used on websites, and wiki presents many of these clustered structures. The on-demand detail in this task is performed through the use of tooltips, which are used to verify involuntary clicks in this structure. Figure 11 shows the comparison between the real scenario and the simulated scenario developed for the tests in this research.

The task applied in the Wiki scenario aims to resemble the same type of actions that the user would perform in a real scenario. In this scenario, there is readable content on the highlighted subject, and the ultimate goal is to find information about specific content in the text and select it. This information is located in a tooltip that has an animation similar to the real scenario and that can cause the MT problem. Figure 12 shows the final state found after executing the task. The sequence necessary for the participant to complete this task is presented below:Interact with PB (disable click);Find the term “Star Wars: Rebels”;Interact with PB (activate the click);Position the cursor on the located component;Click on the generated tooltip.

#### 3.6.4. Scenario 4: Based on E-Commerce Site

The scenario developed based on an e-commerce website was chosen because it simulates a purchase process in e-commerce, which, in turn, uses components such as dynamic menus, which are elements in which, when the cursor is positioned over them, the hierarchies appear on the interface.

These structures bring more dynamism to interaction; however, the waiting time for the menu to be activated and the hierarchies to be displayed can cause MT, especially when using non-conventional interactions to carry out tasks, which can directly compromise the experience of the user of the system.

An important point is that depending on the number of hierarchies that must be accessed in order to reach a certain department within the menus, the possibility of MT occurrence may increase. The use of Dwell Time in this structure can affect the user experience since the time configured for the click to be performed can involuntarily lead to accessing one of the hierarchies of the dynamic menu. Figure 13 shows the comparison between the real scenario and the scenario developed for the tests.

The task in this scenario was designed with the aim of making the participant perform a search and then select a predefined product. Figure 14 shows the final state of the task in the fourth scenario. The sequence necessary for the participant to complete this task is presented below.

Interact with PB (disable clicking);Explore the Hover event in “jewelry and watches”;Locate the items “diamond jewelry”, “sports watches” (men’s watches), and “jewelry packaging and display”;Interact with PB (activate the click);Select the items found.

### 3.7. Evaluation Metrics

For the comparative evaluation between BP and the eViacam software, quantitative and qualitative metrics were selected. Quantitative data were collected by recording the interaction log of each participant during the execution of tasks in the developed scenarios, which are listed below:Total test time;Task completion time using BP and eViacam software;Number of clicks to complete the task using BP and eViacam software;Number of suppressed clicks;Number of clicks on the interface content.

The error in this work was defined as an involuntary click on the interface; for example, a participant when watching a video makes a click and pauses it, without having this objective, thus characterizing an MT in the interface. However, as the task does not have only one path to its solution, the number of clicks necessary to complete a given task will be described in the Second Test Stage section and may be variable, with this number of errors being calculated from the formula presented below.



**NUMBER OF ERRORS = NUMBER OF CLICKS ON THE TASK − NUMBER OF CLICKS REQUIRED**



The time, the number of errors, and the task completion rate of the sets HT+PB and HT+eViacam, in relation to the tasks performed, allow comparisons between the efficiency and effectiveness of each set.

In this article, some aspects of acceptability [50] of the solutions used are considered for the qualitative analysis of this research, namely Aesthetics, Ease of Use, and Usefulness, in addition to aspects of User Interaction Performance [60], Perceptual Effort, Physical Response Effort, and User Satisfaction.

In order to carry out the qualitative analysis, post-test questionnaires were used, with the participants having he option to verbally comment on their respective answers. All questions in the questionnaires used a 5-point Likert Scale (1-worst grade, 5-best grade), in addition to of other features, such as voice recording of the participants and face capture through a webcam.

Each questionnaire was applied at the end of each of the tasks of each of the developed scenarios, with the aim of not losing comments about what was performed in the interface. The audio recorded using a lapel microphone aimed to obtain more detailed data for the qualitative evaluation with a more specific focus on the user experience in relation to the solutions used.

The questions were formulated from the evaluation of a tool, with different questions, which were obtained from different sources [50,60,61,62], having been restructured for the context of PB and the eViacam software.

The organization of the questions took place according to each of the scenarios developed, and the same questions were used for the tests performed for both sets (HT+PB and HT+eViacam), with a comparison being made according to the content of the question and adapting the necessary terms for each of the sets. Below is a list of questions asked.

#### 3.7.1. Scenario 1—Infovis Dashboard

How do you evaluate the “HT+PB or HT+eViacam” system performing interaction on a Map?How do you rate accuracy when using HT on a Map?How do you evaluate the insertion of “PB or eViacam” in the interface, when using the HT?How do you rate the support that “PB or eViacam” offers on the Midas Touch problem with the map?

#### 3.7.2. Scenario 2—Video Plataform

How do you evaluate the “HT+PB or HT+eViacam” system performing interaction in a miniature video?How do you evaluate the “HTRC+PB or HT+eViacam” system performing interaction in a video in full screen mode?How do you evaluate the “HT+PB or HT+eViacam” system performing the configuration in a video in full screen mode?How do you rate the accuracy of HT when using it in a video?How do you evaluate the support that “PB or eViacam” offers in the MT problem for the interaction as a video player?How do you consider interacting with “PB or eViacam” with an on-screen video?How do you rate the insertion of “from PB or from eViacam” in the interface when using the HT?

#### 3.7.3. Scenario 3—Wiki

How do you rate the “HT+PB or HT+eViacam” system for hyperlink interaction?How do you evaluate the “HT+PB or HT+eViacam” system for interaction with the Tooltip generated by the hyperlink?Based on the “HT+PB or HT+eViacam” system, how do you rate the accuracy of the interaction with the hyperlink?How do you rate the support that “PB or eViacam” provides for the MT problem for the hyperlink?In the Wiki scenario there are many components close to interact, so how do you evaluate the support that the “HT+PB or HT+eViacam” system offers for MT prevention?How do you evaluate the physical effort in using the “HT+PB or HT+eViacam” system in this scenario?How do you rate the learning effort of interacting with “PB or eViacam”?

#### 3.7.4. Scenario 4—E-Commerce

How do you evaluate the “HT+PB or HT+eViacam” system using dynamic menus?How do you evaluate the “HT+PB or HT+eViacam” system to carry out the selection of existing components in the dynamic menu?How do you rate navigation with HT on a page that contains a dynamic menu?The dynamic menu has very close components, which interact with cursor positioning. Therefore, how do you evaluate the “HT+PB or HT+eViacam” set to prevent MT in this context?How do you evaluate the physical effort in using the “HT+PB or HT+eViacam” system in this scenario?How do you rate the learning effort of interacting with “PB or eViacam”?

### 3.8. Hypotheses

From the focus of this study, taking into account all the proposed analysis scenarios and based on the information extracted from the collected data, the hypotheses described below were prepared and submitted to a measurement process.

**H1—PB helps the task to be performed more efficiently, in relation to the number of interactions**. Efficiency [63], in this work, is defined in terms of the least number of clicks needed to complete a task. This hypothesis establishes that PB can bring a better level of efficiency to the completion of the task, since this hypothesis assumes that with the use of PB, the proposed tasks can be performed with a smaller number of clicks, causing a reduction in the possibility of interaction errors and involuntary clicks that may affect task completion.**H2—PB enables a reduction in task completion time when compared to the eViacam software**. Focusing on the resolution time of the proposed task, this hypothesis seeks to verify that from the use of the BP, the resolution time of the proposed tasks will be shorter than when compared with the use of the eViacam software, since the difference in time between the groups [64] can show important features that highlight what can be improved between techniques. To confirm this hypothesis, characteristics such as positioning, activation time, and size were considered.**H3—BP’s affordance level is higher than that of the eViacam software**. The use of a component complements its physical characteristics, showing what can be performed with it [62]. The perception for its use was one of the points to verify the need to use BP in the interface. To identify the affordance that BP has compared to the eviaCam component, we sought to analyze the number of element activations among the group participants.**H4—User experience is better with BP than with eViacam software**. The implementation of a technique requires a sequence of evaluations, where it is possible to obtain different feedback from users and thus achieve the best result. Based on the same strategy, this work seeks to identify feedback, using one of the criteria that demonstrate the quality of an application [65], relating the interaction and interface provided to the user experience [19]. It is measured through the analysis of data collected from the questionnaires applied after each scenario and is measured through questions organized on a 5-level Likert scale and directed to the interface element and the system that was offered for interaction with the interface.

## 4. Results

In this section, the results obtained from the analyses carried out on the collected quantitative data are presented, organized according to the order of the hypotheses presented, using the following elements: number of total clicks to complete each task, which was subdivided into suppressed clicks and normal clicks; number of activations; completion time of each of the tasks; and the answers extracted from the applied questionnaires.

Each figure in this section is organized according to the sequence of tests performed: 1—InfoVis Dashboard Scenario; 2—Video Platform Scenario; 3—Wiki Scenario; and 4—E-commerce Scenario. The graphs presented were divided into two groups: BP group, in which the participants performed the tests using BP; and the EV group, in which the participants performed the tests using the eViacam component in the interface. The existence of statistically significant difference between the two groups is represented by an asterisk (*). For this analysis, the two-side *t*-test for independent samples (with *p* < 0.05) was applied.

### 4.1. Efficiency through Interactions

As a basis for the analysis, Table 1 presents the minimum number of clicks needed to complete each of the proposed tasks in each of the developed scenarios, with emphasis on the task in scenario 1 (InfoVis Dashboard), which presents the highest number of clicks needed to complete the task, and scenario 3 (Wiki), which requires at least one click to complete it. Table 4 presents the minimum number of clicks needed to complete each task.

Figure 15 presents a stacked bar chart showing the average of total clicks to complete the tasks and is separated into normal clicks in the interface (blue bar), and the clicks that were suppressed (gray bar); that is, clicks in which the MT was avoided. Note that scenario 3 is the one with the highest average number of clicks to complete the task.

Figure 16 shows the number of clicks on the interface; in all tasks, this number does not show a significant difference. On average, the number of normal clicks was slightly higher in tests performed with PB than in tests using EV, but there was no significant difference between groups.

Thus, for both groups, the number of minimum clicks on the interface to complete the task was similar. The mean number of clicks in scenario 1 (InfoVis Dashboard) in the PB group was approximately 10 clicks, while in the EV, group it was 9 clicks. In scenario 2 (Video Platform) for both groups, the minimum amount was approximately 10 clicks. In scenario 3 (Wiki), the average number of minimum clicks to complete the task was five clicks for both groups. Finally, in scenario 4 (E-commerce), both groups had an average of the minimum number of clicks to complete the task of approximately five clicks.

### 4.2. Task Completion Time

Figure 17 shows the average time to solve the tasks performed by each of the groups (PB and EV). There was no significant difference between the groups in any of the tasks, even though in only one of the scenarios (InfoVis Dashboard) the resolution time for the PB group was greater for completing the task. In the other scenarios, the PB group had a shorter resolution time than the EV group, which may have occurred because the PB had MT prevention as the basis of its operation.

Figure 17, it is also possible to observe some points referring to the confidence interval of scenario 3; in the EV, group a larger interval is observed than the one presented in the other bars of the graph, thus presenting greater total time variations. Scenario 2 (Video Platform) was the one in which the proposed task took the longest to complete, with an average of 94 s for the EV group and 87 s for the PB group.

### 4.3. Level Affordance

Figure 18 shows the number of activations performed by participants in all scenarios. After this analysis, it is important to emphasize that the eViacam component was only activated in scenario 2 (Video Platform), while PB was used by the participants in all proposed scenarios.

The analysis of the number of activations found that there was a difference in significance between the groups in all proposed scenarios, particularly scenario 2 (Video Platform), which was the only one where the use of the eViacam component occurred, highlighting together the confidence interval of the PB group in scenarios 1 (InfoVis Dashboard) and 3 (Wiki), where the variance of activations was high and ended up generating anomalies in the analyzed data. Scenario 4 (E-commerce) had the lowest number of PB activations, with an average of 0.5 activations per participant.

Figure 19 shows the data regarding the average number of clicks suppressed for each scenario. It should be noted that due to the fact that the eViacam component was not activated in scenarios 1, 3 or 4, there is no record of suppressed clicks for the EV group.

The confidence interval in scenarios 1 and 3 continues to be highlighted, as there is a lot of variation in these clicks per user. This is because, all scenarios, the difference in the suppressed clicks of the PB group had a significant difference when compared with the suppressed clicks of the EV group. Finally, it should be noted that, similar to the results presented regarding the number of activations, the average number of suppressed clicks in scenario 4 was small, namely 0.75 suppressed clicks in the PB group.

### 4.4. User Experience

Figure 20 shows the average of the answers of the questions that had a difference in significance, as these bring the context for the discussion of the results; this difference is arranged between the PB and EV groups, with a graph organized according to the list of questions presented in the Section 3.7.

In the answers related to scenario 1, question 2 stands out, where the PB group presented a significant difference when compared to the EV group. For the other questions, even though the PB group presented a higher mean than the EV group, there was no difference in significance between both. When observing the behavior of the respective confidence intervals, in both groups, no accentuated variations were observed. The mean response of the PB group to question 2 was 4.1 and that of the EV group was 3.4.

In scenario 2, none of the response means showed a significant difference between the PB and EV groups (this is why it is not shown in Figure 20). In this scenario, question 4 (Figure 21) stands out, the only one in which the EV group had a higher average than the PB group; however, when observing the confidence intervals, a marked similarity was noted in the average of the responses of both questions. For the other questions, the PB group had an advantage, highlighting the average in question 6 (Figure 21), where the variation was small, demonstrating a common answer among the participants.

In scenario 3, questions 1 and 5 stand out, which show significant differences between the PB and EV groups. Considering the average of question 1, the responses of both groups are similar, with the PB group obtaining an average of 4.8 and the EV group an average of 4.25, but the variety of responses among the participants can be considered small in the group PB and higher in the EV group, a factor that implies the existence of a more evident pattern of answers among the participants of the PB group.

In question 5, a more accentuated difference between the averages can be observed, with the value of the PB group being 4,7 and that of the EV group 3,5, and, as in question 1, the variation is small when comparing both groups. In this scenario, questions 6 and 7 (Figure 21) stand out, where the EV group presented a slightly higher average than the PB group; in question 7, the variance of the answers was greater, presenting means very close to 4,2 for BP and 4 for the EV group.

In scenario 4, the highest number of significant differences was found between the averages of responses between the groups, which were observed in questions 1, 2, and 4 (Figure 20). Questions 1 and 2 have similar behavior, in which the PB group obtained means of 4.5 and 4.4, respectively, and the EV group obtained values of 3.8 and 3.6. Question 4 was the one that showed the greatest difference between the groups, with the mean of the PB group being 4.3 and the EV group of 2.7. Analysis of the other questions shows that the means are similar, and the PB group obtained slightly different mean values. higher than those of the EV group. As feedback and to illustrate the behavior of all the questions asked to the participants, Figure 21 shows all the average values of the participants’ answers in each scenario.

## 5. Discussion

In this section, analyses involving the proposed hypotheses are addressed, together with the definition of characteristics of the collected data that may or may not corroborate them, in order to identify points in which the data may present similarities and allow the extraction of qualitative aspects that allow the assessment of the hypotheses of this study.

### 5.1. Efficiency through Interactions

Hypothesis 1, which aims to highlight the efficiency of the PB, is discussed in Figure 16 and Figure 17, showing, respectively, the total number of clicks to perform the task and the number of normal clicks in the interface. The groups of participants do not present a significant difference from the extracted results, and the minimum number of clicks to complete each of the tasks is similar, a fact that does not help to confirm the hypothesis in question.

However, based on the analysis of the answers to question 2 of scenario 1, this contradicts the initial perception, since there was a difference in significance between the responses of the groups of participants, and the PB group registered a superiority in the efficiency of the solution proposed in this study to mitigate the MT.

Some qualitative data referring to the participants’ audio recordings during the tests were important for the analysis of hypothesis 1. One of the participants in the PB group stated that Pactolo Bar makes the activation and deactivation process faster—*“The Pactolo Bar is faster to enter”*— that, although the collected data indicate equivalence in the minimum number of clicks to complete the task, it can be observed that the interaction with the PB is a more frequent act. Furthermore, they stated that, even so, the users completed the task with a number of clicks similar to participants in the EV group.

As for the EV group, in general, the participants registered a level of difficulty in interacting with the eViacam component, a difficulty linked to its size and position in the interface. Freitas et al. [41] identified in their suggestions that when interacting with head tracking, it is advisable for the graphic elements to be centered on the screen. Another important point is that the act of not deactivating the involuntary click of the eViacam component leaves the participant more focused on the interaction task to be performed but increases the possibility of MT.

### 5.2. Task Completion Time

Time was one of the characteristics evaluated in the collected data and one of the hypotheses evaluated in the process. As shown in Figure 18, the times to complete the tasks did not differ in significance, and in tasks 2, 3, and 4, the time to complete tasks in the PB group was slightly lower than in the EV group. However, when carrying out the analysis of the participants’ reports, a certain fear in use of the eViacam component was perceived, as observed in Figure 19, which presents the number of activations of both techniques. Thus, if the participants had used the eViacam component more frequently, it is likely that the time element would be more relevant to the analysis.

The interaction action from the use of the eViacam component proves to be a factor that reduces the average speed for the interaction actions. Some users reported during the questionnaire response that *“interacting with PB is fast, it makes me want to use the bar”*. While other users did not try to use the eViacam component, as it would visibly affect the completion of the task; for example, one of the participants, when using the eViacam component in the Video Platform task, and when asked about the reason for not having used the component, reported that *“Yes, I did not use it because I knew I could miss the click when I was going to click”*.

### 5.3. Level Affordance

Hypothesis 3 was related to the affordance factor that PB could bring to the participants. This factor refers to the use of PB as a method to avoid the MT problem in the proposed tasks; the number of PB activations (Figure 19) presents PB as the most used technique in the scenarios, and its use obtained a greater difference in significance compared with the EV group.

It is evident that there is a need to use a MT prevention method; in this sense, scenario 2 reflects this need, as both techniques were used, with some participants highlighting that *“Here, I think I will need to use the bar”*, in addition to the fact that in the other scenarios, there was no use of the eViacam component.

The minimal use of the eViacam component can be sharply justified based on some comments made by the participants, such as *“I did not use it because I found it difficult to go there!”*; *“I forgot the button upstairs”; “His position makes me uncomfortable"*; and *“I have to move my head a lot to get there!”*. These responses demonstrate the users’ concerns in using the eViacam component in factors, such as interaction difficulties, positioning, and tracking failure, among others.

Overall, the perception that was extracted from the participants’ comments is that the use of the eViacam component was impaired due to elements such as its aesthetics, position, and interaction. This is because, when compared to PB, this is activated and deactivated through the Hover action. An interesting point that can be observed was from the speech analysis of one of the participants, who said, *“I forgot the button up there”*.

The participant’s above quotation clearly characterizes the results obtained in relation to the number of suppressed clicks, which are presented in Figure 20, from which it can be objectively observed that the PB was more used and had a greater number of suppressed clicks in the interface.

Since the suppressed clicks, in general, brought the user more freedom to perform the proposed task, this behavior can be evidenced in scenario 3, where the confidence interval present in the graph presents a very diverse behavior in relation to the suppressed clicks.

The freedom that the user has to perform the task reflects the question asked after scenario 3 was performed, and the results can be seen in the data shown in Figure 21. In this scenario, participants were asked about how the support offered by PB is evaluated in the context of MT prevention.

The responses to these questions show that the more frequent use of PB offers greater flexibility for the participant to answer questions about the proposed solution, as can be seen in the following responses from participants about the eViacam component:*“I did not use the button so I ca not say”* and *“I did not even remember him!”*. The analysis carried out for this question leads to the conclusion that BP can be considered a good solution for the prevention of MT. BP presented good results in its composition, and its level of affordance satisfied its need for use. Without the help of Dwell Time to interact with the graphic element, the results show, qualitatively and qualitatively, that the use of Dwell Time can affect task performance. For instance, H. et al. [39] show that this technique distracts the user and that errors occur in the proposed task.

According to answers to this question, the more frequent use of PB offers greater flexibility for the participant to answer about the proposed solution, as can be seen in the following responses from participants about the eViacam component: *“I did not use the button so I ca not say”* and *“I did not even remember him!”*. The analysis carried out in question leads to the conclusion that BP can be considered a good solution for the prevention of MT.

### 5.4. User Experience

Hypothesis 4 addresses some points that highlight the use of PB in comparison to the eViacam component in the interface. Some of the participants’ statements help to describe how PB obtained a significant difference in the results obtained.

The accuracy of the HT+PB set in an interface that uses a map is demonstrated in the answers to question 2 of scenario 1, which are shown in Figure 19. It is clear that the use of the eViacam component in this scenario could generate more involuntary clicks, which can be seen from the following response from one of the participants:*“If I were to click there, I would concentrate on clicking and I would lose the map!”*.

In general, analyzing the answers of the questionnaires answered by the participants shows that the interaction with the PB is faster and the precision for the interaction is more accentuated. This can be considered due to the fact that the PB is a component that has a size that brings more security so that the user can access it without fear of making a mistake when activating or deactivating it.

Scenario 3 adds two issues that are important for analyzing the use of PB in the interface. The first is question 1, which refers to the use of the hyperlink in the system used by the participants, and the second is question 5, which refers to the support that PB offers to perform the interaction with the hyperlink.

In question 1, a significant difference was observed between the groups of participants, a fact that is evidenced by the non-use of the eViacam component by the participants during the tests (Figure 19). When analyzing the collected reports, one of the responses that can be highlighted is *“In this scenario, PB is important!”*. This comment suggests the need for a method to avoid the MT, and this scenario was the one that had the most variation in its data regarding the number of suppressed clicks.

Question 5, on the other hand, demonstrates that scenario 3 is a favorable scenario for the use of PB, since it is a scenario where the interface elements that use interaction are very close and the action that uses the tooltip (action that leads to another window) demands a certain time interval, which can cause the MT. However, in general, the analysis of the responses referring to scenario 3 leads to the conclusion that the participants varied their strategy a lot, where some activated the PB for a long time and others did not.

From the analysis of the answers referring to scenario 4, but specifically the answers to question 1, the difference in significance in the results can be observed, raising the question about the use of the HT+PB system in interfaces that have the dynamic menu element.

It can be stated empirically that the use of the dynamic menu is very intuitive and fast, but when using an unconventional interaction in the system, the MT may arise. When analyzing the participants’ responses, a positive rating can be seen in the interaction with the PB, as it presents a more pronounced level of accessibility to the participant, as well as its positioning in the interface and its ease of being activated and deactivated, which favor the interaction with the dynamic menu feature.

In some cases, it can be seen that the participants activated the PB to observe and memorize where the components were on the menu, as seen in the response of one of the participants who issued the following observation: *“I thought it was very good, I was able to use the PB and look at the items to click”*. Question 2, still from scenario 4, addresses the selection interaction of the components present in the dynamic menu. Thus, it is possible to analyze the participants’ experience when carrying out the proposed tasks.

At this point, selection is one of the principles of interaction with the elements of the graphical interface, and this factor is important for a good user experience. Question 2 obtained a significant difference between the groups of participants, making it possible to show that the use of the HT+PB system offers better support so that the participants can interact with the interface elements without causing MT.

When observing the answers referring to question 4 (scenario 4), it can be concluded that the users’ experience proved to be better for those who were able to perform the proposed tasks using PB. Participants were asked about the overview the site provides and how PB can help them accomplish the task. From the results presented in Figure 21, it is clear that the participants had the perception that the PB offers better support for the use of the dynamic menu.

Although in many cases the HT affected the performance of the test, it could be observed that some users used the PB as a security feature to avoid possible errors caused by the HT. This can be considered in light of the fact that, when activating the PB, it does not run the risk of performing some action that would affect the interface configuration (TM prevention). Performance can be improved by adding another unconventional interaction. Menna et al. [33] clarify that the use of these different interactions in a system is the key to usability by the user. Thus, BP can be adapted to other means of interaction.

## 6. Conclusions

In this section, an attempt is made to summarize the main points that were found during the development of this research, taking into account the positive and negative aspects involving the results obtained and the proposed hypotheses, below is the list of the main discoveries.

The best position to place the BP is on the left of the screen, with a possible secondary position on the top of the screen.The data collected both quantitatively and qualitatively demonstrate, clearly, the discussion that can be obtained regarding the hypotheses generated.H1 and H2 did not present results with a difference in significance between the data. However, they present advantages in the use of BP in the developed scenarios.H3 demonstrates that the BP affordance is greater than that of the eViacam component, making it possible to improve the BP’s characteristics and incorporate new functions that can help in better performance.The results obtained demonstrate the measurement of H4, where the participant’s UX was better using BP.From the analysis of the collected data, it can be observed that the UX of the participants who used the eViacam component was impaired due to elements such as positioning, size, form of interaction, and aspects of perception.BP provides more stability for interaction, being easier to access, activate, and deactivate;Scenarios that have many clustered elements, such as Video Platform and Wiki, demonstrate the greater efficiency of BP in preventing MD.

This work brought the aspect of user experience in the context of using non-conventional interaction in web content, aiming to alleviate the Midas Touch problem through a graphic element inserted in interface, called Pactolo Bar (PB). Some topics of studies focused on the scenario presented; for example, the lack of studies on user experience, interface aspects, and comparisons of techniques demonstrate a lack of studies that can help in development technologies incorporated into the end user’s environment.

Through the elaborate study, tests, and collection and analysis of the data, we reached the conclusion of the beneficiaries of the presented work, namely that the users do not present the problem of the Midas Touch during their interaction, or at least they can reduce their occurrence. This benefit occurs through the improvement of user experience, highlighting the level of affordance of the proposed technique and important characteristics such as positioning, interaction event (through Hover), and shape. Finally, non-conventional interaction methods receiving improvements, as technology evolves and adapt according to the studies used in different scenarios in which they can be inserted.

Some isolates were found during the evolution of the article, such as the number of test participants, although it is shown in the literature that when 12 [52] participants present data for analysis, this number does not represent the population as a whole. Another limiting aspect of the research is in the tests with participants who use these technologies for interaction, such as quadriplegics or people with paralysis of the upper limbs; this aspect of the research can bring more detailed data to support the conclusion.

The use of BP in other scenarios can bring different views of what was collected, which is a limitation of the research developed. These new scenarios, such as navigating file systems, present other tasks that can be fitted with the routines of participants, such as copying and pasting and dragging and dropping. Another limiting aspect of the research is related to the scale of the BP, such that it is possible to apply other functions in other scenarios, and not only in activating and deactivating the click. However, these extra functions also can improve the user experience in a complex task to be completed.

## 7. Future Works

From what was presented with the results obtained, it is possible to indicate some future works that can analyze new methods to prevent MT when using non-conventional interactions, in addition to other types of data that may be collected and that may improve the conclusion about the developed techniques, such as eye tracking; and sentiment analysis.

Another important point is the verification of the insertion of the PB context in different environments, such as smart TV software; smartphones, and large displays, among others. Leaving the field of digital applications, MT prevention methods can be applied to physical visualization techniques, as non-conventional interactions bring new scenarios to the use of technologies, mainly in computational environments in which, as seen, they can enable the occurrence of MT, thus generating new testing and analysis scenarios.

New strategies can be generated with the objective of mitigating the MT, making it possible to use intelligent algorithms that seek to analyze the user’s behavior and changes in the runtime of the interface, so that it can be adapted to the type of unconventional interaction used by the user.

Other works that do not involve exclusive techniques can seek to mitigate MT, such as research on the use of non-conventional interactions in interfaces that are not adaptable, which seek to develop methods that demonstrate how the interface can be organized for their use, and focus on components that use interactions and visualization techniques in general.

## Figures and Tables

**Figure 1 sensors-23-02110-f001:**
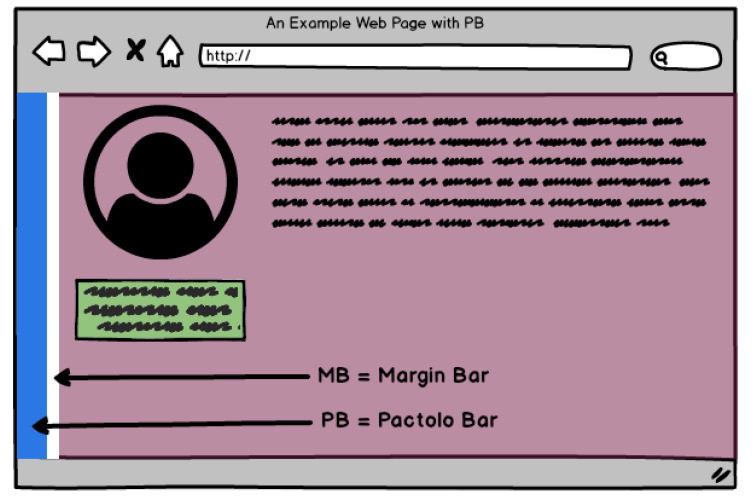
Representation of the Pactolo Bar running in a web page.

**Figure 2 sensors-23-02110-f002:**
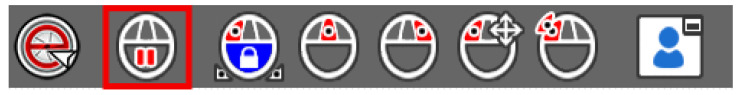
Software eViacam Cursor Events Toolbar.

**Figure 3 sensors-23-02110-f003:**
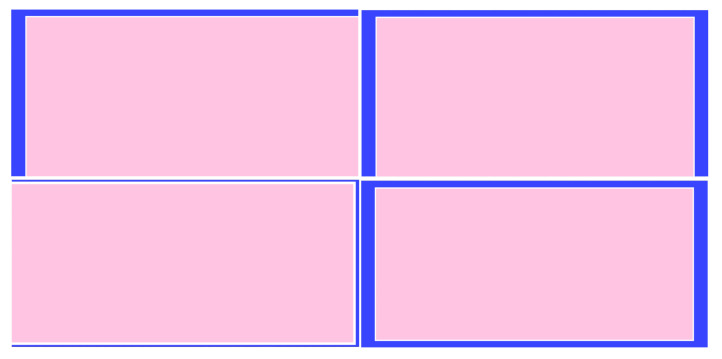
Examples of random bar positions.

**Figure 4 sensors-23-02110-f004:**
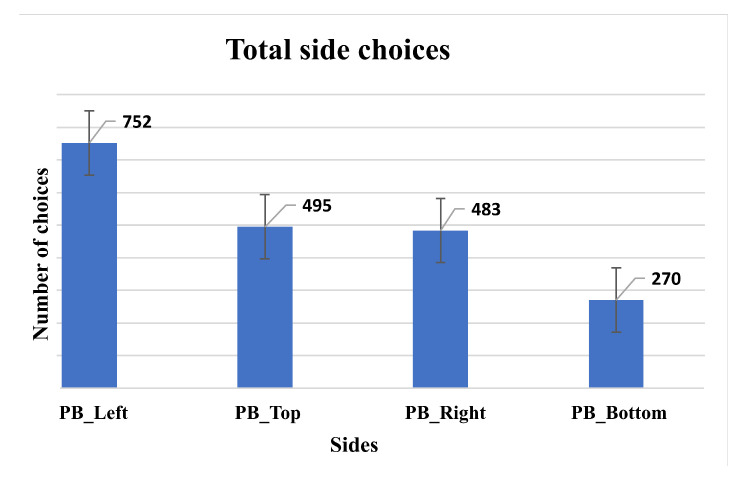
Preferences of users during the first stage of tests.

**Figure 5 sensors-23-02110-f005:**
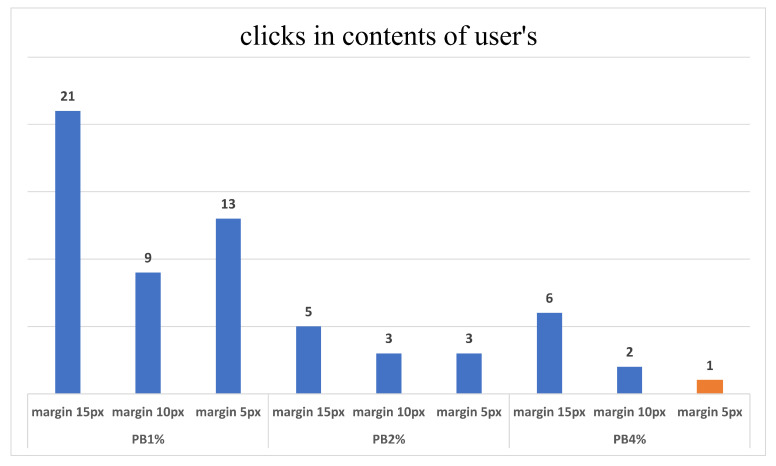
Amount of wrong clicks in synthetic scenarios.

**Figure 6 sensors-23-02110-f006:**
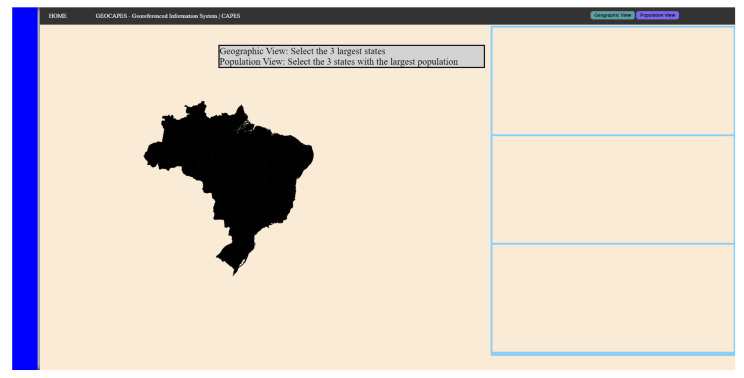
Simulated scenario 1 based on a generic dashboard in a web service with PB.

**Figure 7 sensors-23-02110-f007:**
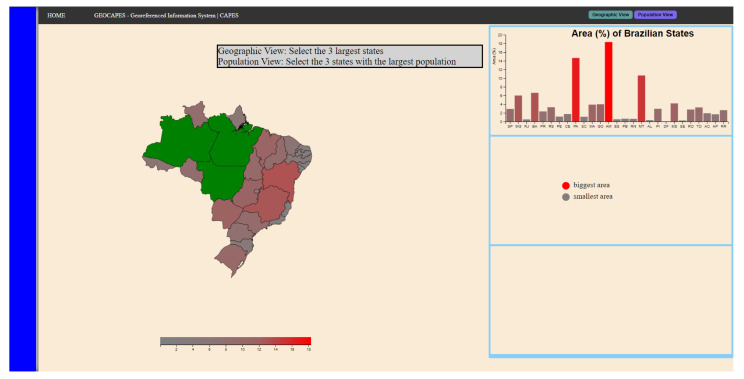
Final state of the first task step in scenario 1.

**Figure 8 sensors-23-02110-f008:**
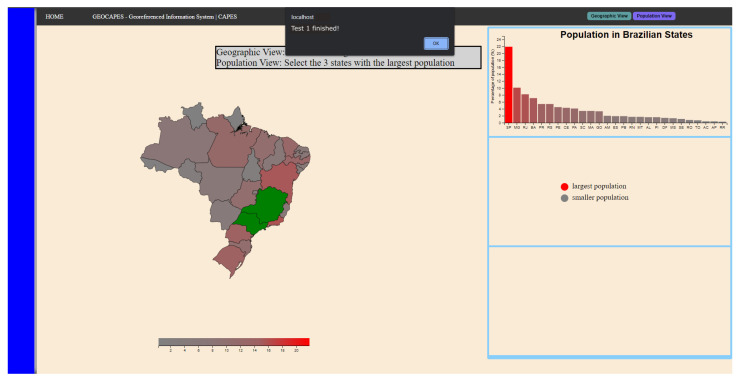
Final state of the second step of the task in scenario 1.

**Figure 9 sensors-23-02110-f009:**
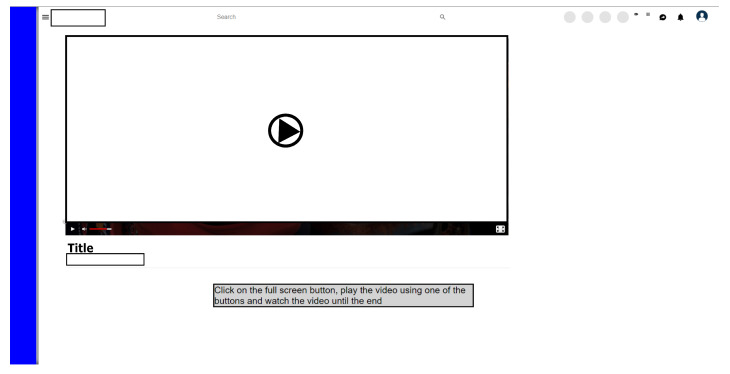
Simulated scenario 2 based on a generic video platform in a web service with PB.

**Figure 10 sensors-23-02110-f010:**
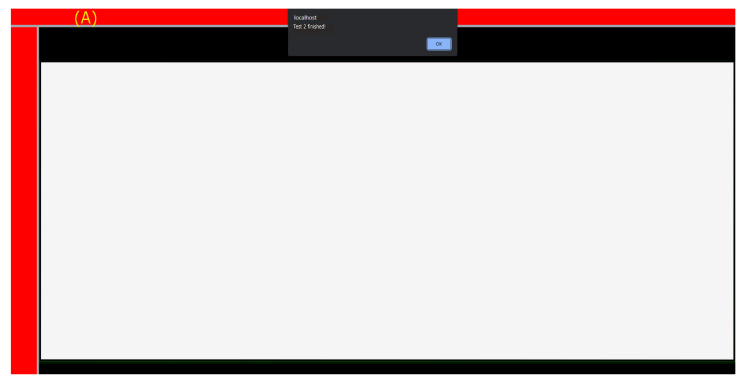
Final state of the task in scenario 2 in A secondary PB.

**Figure 11 sensors-23-02110-f011:**
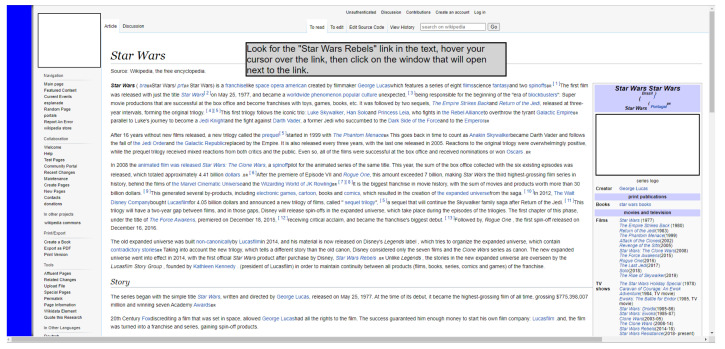
Simulated scenario 3 based on a generic wiki platform in the web service with PB [59].

**Figure 12 sensors-23-02110-f012:**
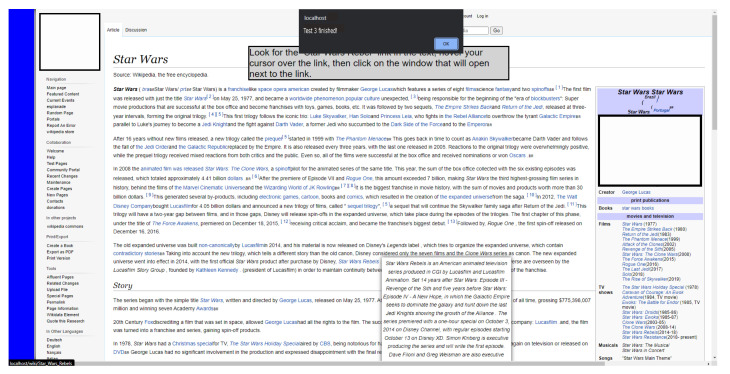
Final state of the task in scenario 3 [59].

**Figure 13 sensors-23-02110-f013:**
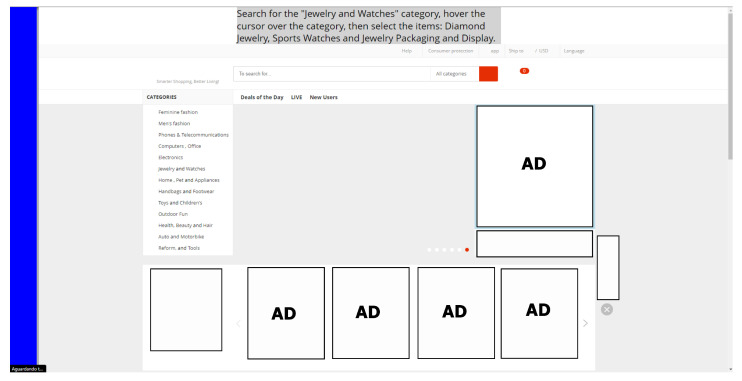
Simulated scenario 4 based on an e-commerce platform in a web service with PB.

**Figure 14 sensors-23-02110-f014:**
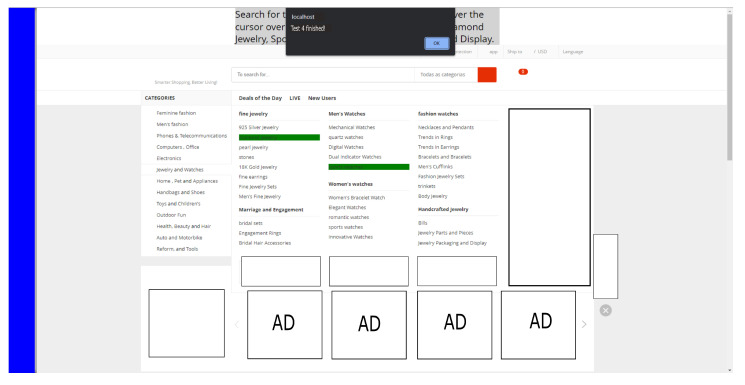
Final state of the task in scenario 4.

**Figure 15 sensors-23-02110-f015:**
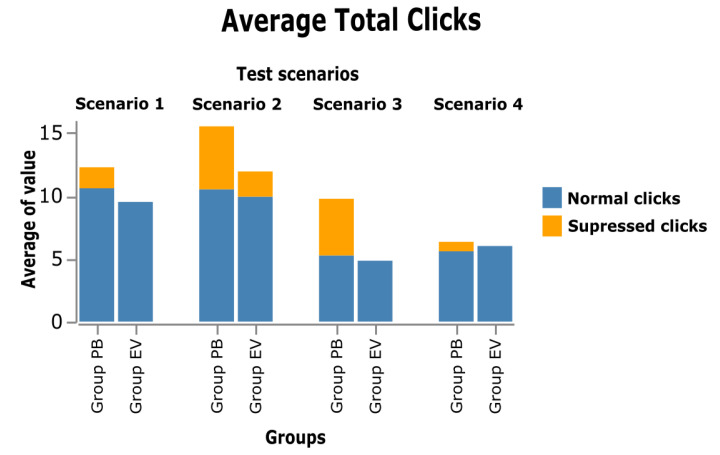
Average clicks to complete the task.

**Figure 16 sensors-23-02110-f016:**
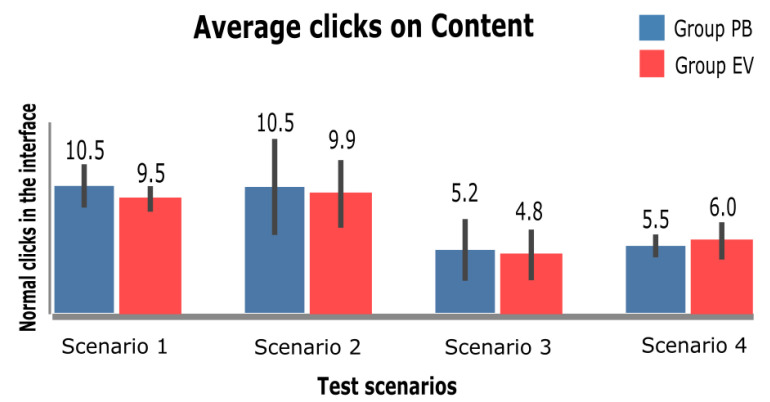
Average of normal clicks on the interface.

**Figure 17 sensors-23-02110-f017:**
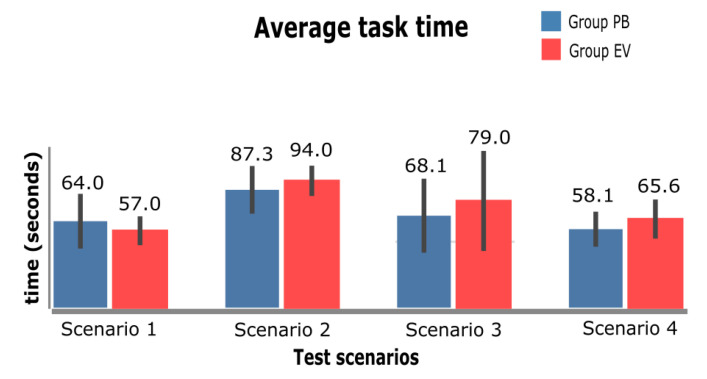
Average time to complete tasks.

**Figure 18 sensors-23-02110-f018:**
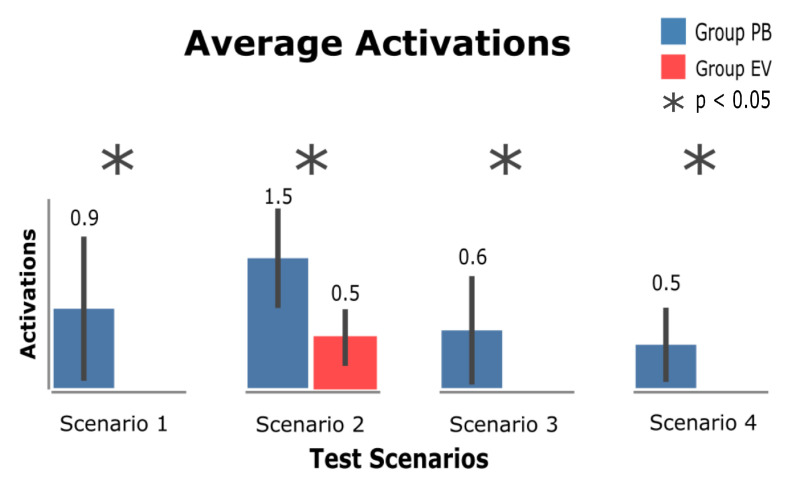
Number of BP and eViacam component activations.

**Figure 19 sensors-23-02110-f019:**
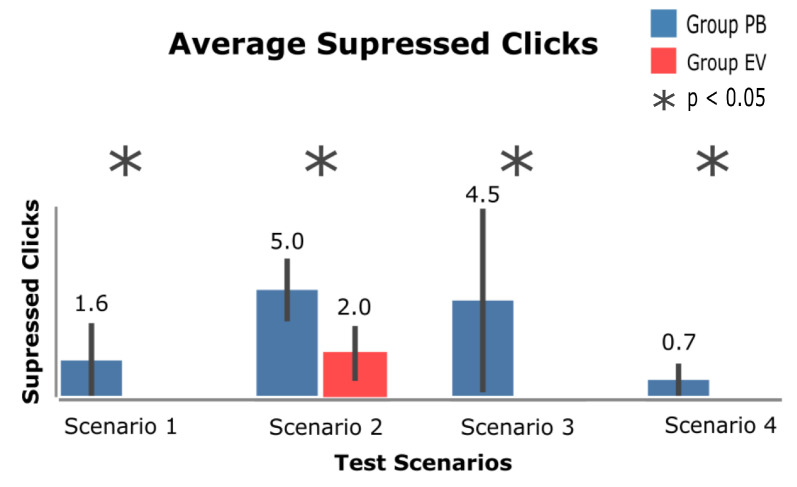
Average clicks suppressed across groups on each task.

**Figure 20 sensors-23-02110-f020:**
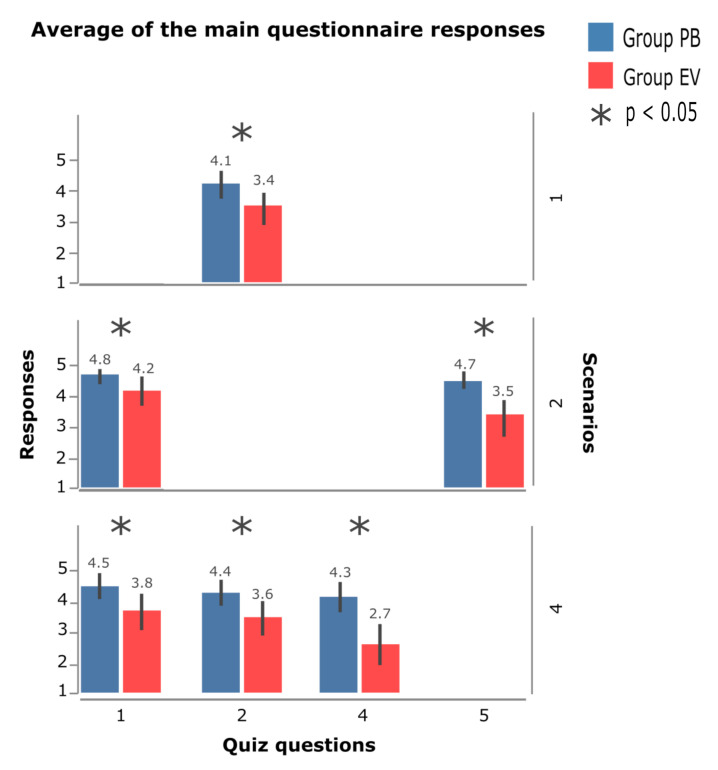
Mean of questionnaire responses between groups.

**Figure 21 sensors-23-02110-f021:**
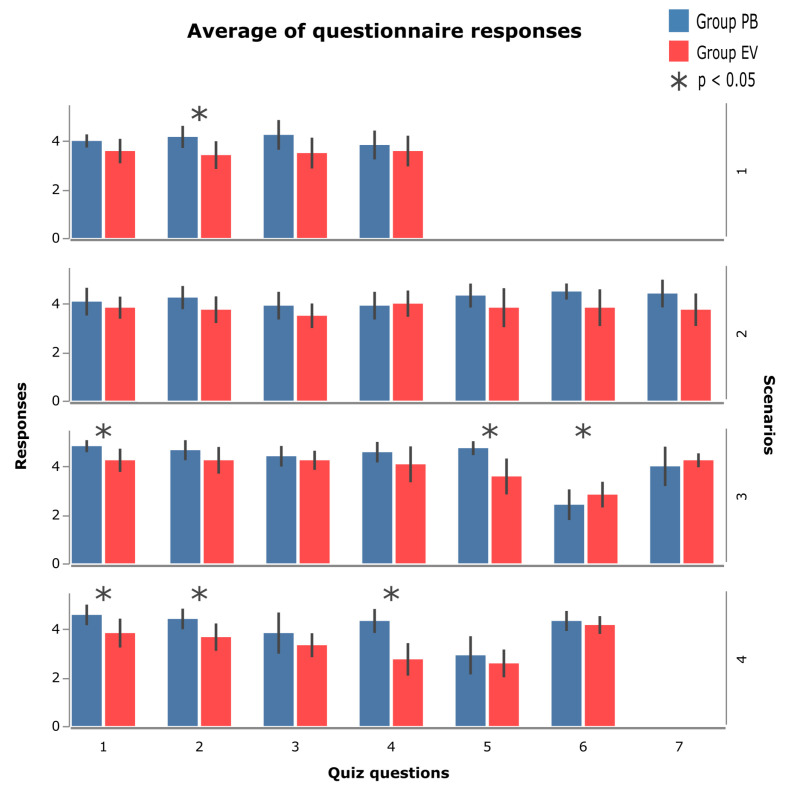
Average of all questionnaire responses.

**Table 1 sensors-23-02110-t001:** Characteristics of participants in the first stage of tests.

Characteristic	Quantity
Sex	Male: 12 | Female: 8 (participants)
Age	Between 18 and 40 years old
Daily use of computers	all participants
Use of web browsers	all participants
Know Non-Conventional Interactions	Yes: 12 | No: 8 (participants)
Used HT	5 participants
Used other Non-Conventional Interactions	7 participants

**Table 2 sensors-23-02110-t002:** Characteristics of participants in the first stage of tests.

Elements	Values
Sex	Male: 16 | Female: 8 (participants)
Age	Between 15 and 40 years old
Daily use of computers	all participants
Commonly used systems	Web browser, Video platform, Social Midia, Text editors and File manager: $\overset{̰}{1}00$% | Video/photo editors: $\overset{̰}{3}3$% | Specific computational systems: $\overset{̰}{2}5$% (participants)
Know Non Conventional Interactions	Yes: 20 | No: 4 (participants)
Used HT	3 (participants)
Used other Non Conventional Interactions	Eye tracking: 1 | Gestures: 7 | Voice: 16 and Triggers: 2 (participants)

**Table 3 sensors-23-02110-t003:** Elements and values used in the first stage of tests.

Elements	Values
Number of Bars on the Screen	2, 3, or 4
Position on the Screen (PB and MB)	Left, Right, Up, and Down
Orientation on the Screen (PB and MB)	Horizontal and Vertical
Size of the PB	1%, 2%, and 4%
Size of the MB	5px, 10px, and 15px

**Table 4 sensors-23-02110-t004:** Minimum number of clicks to complete each task.

Scenario	Number of Clicks
Scenario 1—InfoVis Dashboard	8
Scenario 2—Video Platform	4
Scenario 3—Wiki	1
Scenario 4—E-commerce	3

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
