# Peer review of "Pactolo Bar: An Approach to Mitigate the Midas Touch Problem in Non-Conventional Interaction"

_sensors, 2023, doi:10.3390/s23042110_

Round 1

Reviewer 1 Report

The research analyzed the technical advantages of a visual auxiliary toolbar through a diversified small-scale test, but considering the innovation in technology research and development, the overall lack of necessary technological innovation, it is suggested to consider more comprehensive factors, especially for the practical technical problems, such as:

1. Whether it is necessary to discuss the effect on different display devices rather than purely discuss software issues;

2. The experimenter under test lacks a more detailed explanation. In view of different technical backgrounds and different browsing requirements, it is necessary to carry out scientific measurement on the tool itself, especially the use of a richer multi-group reference model;

3. There are many similar tools or plug-ins. I don't think this research and development has obvious advantages. What are the differences between it and other similar tools?

Author Response

Dear Reviewer.

We thank you for your time dedicated to reviewing the article, and especially for the highlighted points that improve the quality, organization, and clarity that the work seeks to convey. Attached are the answers to each point that was indicated, we analyzed all the points and tried to meet expectations.

Reviewer 2 Report

You have written a quality paper but you need to address some major errors to improve the standard of the paper. I explain my concerns in more detail below. I ask that the authors specifically address each of my comments in their responses.

1.     The benefits of the study to the shareholder should be added at the end of the Abstract section.

2.     The keywords should be a minimum of five keywords.

3.     In the Introduction section, there are many paragraphs. In scientific papers, each paragraph includes a minimum of 5-6 sentences. Thus, the related paragraphs should be combined.

4.     In section “3.2. PROFILES OF PARTICIPANTS”, detailed information about the participants such as experiments, gender, expert area, etc. should be given.

5.     Which criteria are taken care of when the groups create? How did you sure the equality of created groups? How did you select group members? How many groups are there, and what are their aims? etc. all the information about the groups should be explained in detail.

6.     H1 and H2 should be discussed with the related literature, too.

7.     In the “Results” and the “Discussion” sections, in the subtitles, the related/suitable titles should be used instead of H1 or etc.

8.     One of the most important sections of a scientific paper is the Discussion section,  and discussion should be made with other studies with similar and/or contradictory results to the results obtained in the study.

9.     Section 6, “FINAL REMARKS”, this section should be deleted. The idea from this section is already discussed in the Discussion section.

10.  The “CORE FINDINGS” should be changed to “Conclusion”. Moreover, in the “Conclusion” section the author(s) should be summarized only important results and based on this how the study will fill the missing gap in the literature. Moreover, at the end of the Conclusion section, the benefits of the study to whom/how/what should be added.

11.  Also, the limitations of the study should be explained in the paper.

Author Response

Dear Reviewer,

We thank you for your time dedicated to reviewing the article, and especially for the highlighted points that improve the quality, organization, and clarity that the work seeks to convey. Attached are the answers to each point that was indicated, we analyzed all the points and tried to meet expectations.

Round 2

Reviewer 2 Report

Dear author(s),

Thank you for your correction.